# Instruction Vulnerability Prediction for WebAssembly with Semantic Enhanced Code Property Graph

## ABSTRACT

WebAssembly (Wasm) is a universal low-level bytecode designed to build modern web systems. Recent studies have shown that technologies such as voltage scaling and RowHammer attacks are expected to increase the likelihood of bit flips, which may cause unacceptable or catastrophic system failures. This raises concerns about the impact of bit flips on Wasm programs, which run as instructions in web systems, and it is an undeveloped topic since the features of Wasm differ from traditional programs. In this paper, we propose a novel paradigm, namely IVPSEG, to understand the error propagation of bit flips within Wasm programs. Specifically, we first use Large Language Models (LLMs) to automatically extract instruction embeddings containing semantic knowledge of each instruction's context. Then, we exploit these embeddings and program structure (control execution and data transfer) to construct a semantic enhanced code property graph, which implicates the potential path of error propagation. Based on this graph, we utilize graph neural networks and attention diffusion to optimize instruction embeddings by capturing different error propagation patterns for instruction vulnerability prediction. In particular, we build a Wasm compilation and fault generation system to simulate bit flips at Wasm runtime. Our experimental results with 14 benchmark programs and test cases show IVPSEG outperforms the state-of-the-art methods in terms of accuracy (average 13.06%↑ ), F1-score (average 14.93%↑), and model robustness.

## KEYWORDS

WebAssembly, Bit flips, Instruction Vulnerability Prediction, Error Propagation

## 1 INTRODUCTION

WebAssembly (a.k.a., Wasm) is an increasingly important low-level bytecode format with high efficiency and fast execution[24, 38]. It serves as a compilation target for high-level languages such as C/C++, enabling developers to port native programs to the web[19, 37]. And Wasm's native-like performance may transform modern web application development. For example, Figma and Google Earth are prominent examples of applications leveraging Wasm to achieve high performance [2, 23].

Due to different program features (e.g., frequent memory and stack operations, no direct system call), Wasm programs have unique security threats. Recent studies have shown that technologies such as RowHammer attack [31], Dynamic Voltage Frequency Scaling (DVFS) attack [36], and clock glitching [42] are expected to increase the likelihood of bit flips, which may cause unacceptable or catastrophic system failures by changing the memory data or instruction sequence of Wasm programs[4, 32]. For example, as shown in Figure 1(a), bit flips occurring in physical memory or registers may break the integrity of data or code of Wasm programs stored in the memory[55], causing errors in web applications. This

raises concerns about the impact of bit flips on Wasm programs, which is an important and undeveloped topic.

Currently, many methods have been proposed to detect bit flips at the hardware or software level, such as Error-Correcting Codes (ECC)[33]. Still, they cannot completely avoid bit flips[12, 15]. Besides, there is a gap in predicting how instructions may cause program errors when affected by bit flips. Therefore, inspired by Emscripten[53], our work mostly focuses on more fine-grained modeling of Wasm program vulnerabilities at the LLVM[1] instruction level (also known as Instruction Vulnerability Prediction[21, 51]). The most common method is based on hardware fault injection[39, 50], which simulates hardware faults, such as bit flips or memory modifications, and then identifies vulnerable instructions through statistical analysis. However, these methods require full fault injection, and the resource consumption grows exponentially with program size. Thus, to expedite assessments, researchers aim to reduce the number of required fault injections while keeping the accuracy of instruction vulnerability prediction. Unfortunately, there are still several challenges:

**C1: Insufficient structure semantic for modeling the error propagation pattern caused by bit flips.** Due to the mixture of control execution, data transfer, and other structures, the propagation of errors through instructions can be extremely complex during Wasm runtime. As shown in Figure 1(b), the Wasm program ($factorial.wat$) can be interconverted with LLVM ($factorial.ll$). And when the memory data "$local\ get\ 0$" (also expressed as the register "%3" in the instruction " $store\ i32\ \%0, i32 * \%3,\ align\ 4$ ") is corrupted, it will not be detected by web systems but propagated to registers operated by subsequent instructions, such as the path of "%3 → %10 ⋯ → %15", which eventually leads to incorrect returns. However, when the register "%2" in the instruction "%2 = $allocai64, align8$" has an error, it will be masked during propagation. Thus, it is a challenging problem to model the whole path of error propagation, which can provide better interpretability for error analysis, and predict truly vulnerable instructions.

**C2: Lack of instruction semantic for enhancing the Wasm program representation.** Specifically, some studies[34, 52] focus on manually designing heuristic features and predicting instruction vulnerabilities by performing partial fault injection and machine learning. While reducing resource consumption is a notable achievement, these heuristic features do not always correlate strongly with instruction vulnerabilities, especially for Wasm programs. Besides, they do not clarify the importance of data transfer between instructions and the inherent semantics of instructions. For example, the instruction *"%5 = icmp eq i32 %4, 0"* in Figure 1(b), means comparing the result *"%4"* of *"load"* with 0 and storing the result in *"%5"*, which is used by *"br"* instruction. Thus, mining semantic knowledge in context and extracting robust embeddings to represent programs fully are crucial for instruction vulnerability prediction.

---

[1]https://llvm.org/

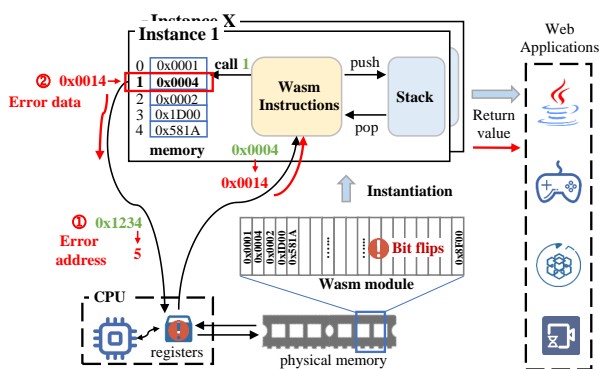

**(a)** The effect of hardware faults on Wasm programs

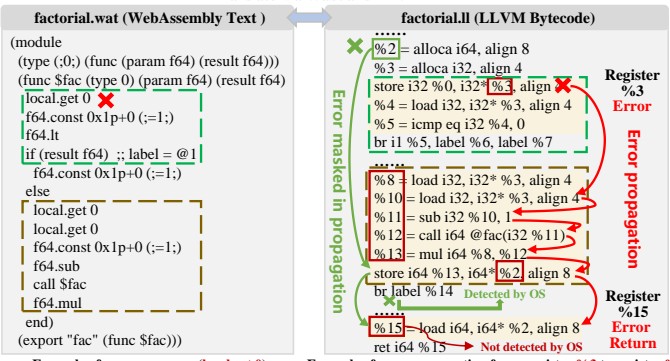

**(b)** Complex propagation of errors at the instruction level

**Figure 1: Schematic diagram of the Wasm program error generation and propagation at the instruction level. As shown in (a), when a bit flip occurs, the original data "0x0004" may become "0x0014" or the original address "0x1234" may become "0x1235", which will cause the Wasm instruction to run incorrectly.**

To address the above challenges, we develop IVPSEG, a novel paradigm for ensuring the security of Wasm runtime against potential bit flips. Specifically, to mine the semantics of instructions in context, we first use the Large Language Model (LLM) to automatically extract the semantic embeddings of instructions. Then, we notice that errors propagate during control execution and data transfer at the Wasm runtime. Thus, we exploit the structure and instruction semantics to construct a semantic-enhanced code property graph, which implicates the potential path of error propagation. In particular, instructions in different basic blocks are distinguished, which can provide more refined information for locating vulnerable instructions. Finally, we utilize graph neural networks (GNNs) and attention diffusion to optimize instruction embeddings by capturing different error propagation patterns. Based on WABT[2] and LLFI[27], we build a Wasm compilation and fault generation system, which compiles Wasm to LLVM intermediate representation (IR) and simulates register or memory bit flips at Wasm runtime, which are carriers for data transfer. The experimental results on the 14 benchmark programs show the effectiveness of IVPSEG compared to the state-of-the-art methods. Our main contributions are as follows:

- To our knowledge, we are the first to study the impact of bit flips on Wasm programs. We propose a novel paradigm for mining error propagation patterns of bit flips by using multi-layer structure semantics and instruction inherent semantics.
- We leverage the latest LLM technology to extract the context of data transfer within Wasm programs. This context helps us enhance the instruction's inherent semantics to understand better how errors are propagated.
- Unlike traditional approaches, our method captures the importance of numerical carriers in data transfer and the hierarchical structure of Wasm programs for enhancing the structure semantics. We also adopt GNNs and attention diffusion to model the error propagation at the instruction level.
- We build a Wasm compilation and fault generation system, which compiles Wasm to IR and performs bit flips during the runtime of Wasm programs. Extensive experiments with 14 benchmark

programs and test cases are conducted to validate the effectiveness of our method. The verifiable data and code are published in https://anonymous.4open.science/r/IVPSEG-9377/[3].

## 2 PRELIMINARIES

We list the main variable notations in Appendix Table 3. Given a Wasm program $S$ (native program or Wasm binary), which can be compiled/decompiled into the IR instruction sequence $\Phi = \{n_1, n_2, ..., n_N\}$, where $N$ is the total number of instructions. The $\Phi$ can be divided into a basic block sequence $\Delta = \{B_1, B_2, ..., B_M\}$, where $M$ is the total number of basic blocks. Each basic block $B_j$ consist of a set of instructions $\{n_i | n_i \in B_j, n_i \in \Phi\}$. Based on the program analyzer, we can obtain the instruction execution process, data dependencies, and semantic text, such as opcodes, operands, and registers. These can be represented as a set of entity-relation-entity triples $T = \{t_1, t_2, ..., t_J\}$ and a set of features $\Lambda = A_{ins} \bigcup A_{bb}$, where $J$ is the total number of relations, $A_{ins} = \{I_1, I_2, ..., I_N\}$ presents the features of instructions and $A_{bb} = \{b_1, b_2, ..., b_M\}$ presents the features of basic blocks. Each triple $t_i$ is in the form of $(C_i, r_i, C_k)$, where $C_i, C_k \in \Phi \bigcup \Delta$ and $r_i$ is the relation between the entities $C_i$ and $C_k$. Based on that notations, we can define instruction vulnerability and formulate the problem of instruction vulnerability prediction as follows:

**Definition 1 (Instruction Vulnerability).** *Instruction vulnerability is the probability that the program results may be incorrectly raised by the change of instruction $n_i$ due to bit flips during execution, denoted as $y_i$. $Y$ is the set of all instruction vulnerabilities in the program $S$.*

**Problem 1 (Instruction Vulnerability Prediction).** *Given a small set of instruction vulnerability $Y_{train}$ inferred by methods like hardware fault injection for training, instruction vulnerability prediction can be formulated as a semi-supervised learning problem:*

$$\{\Phi, \Delta, T, \Lambda, Y_{train}\} \xrightarrow{\mathcal{F}(\cdot)} Y. \tag{1}$$

## 3 METHODOLOGY

In this section, we present the proposed IVPSEG, as shown in Figure 2(a), an intelligent framework for resisting hardware faults,

---

[2]https://github.com/WebAssembly/wabt

[3]The repository is anonymized for peer reviewing.

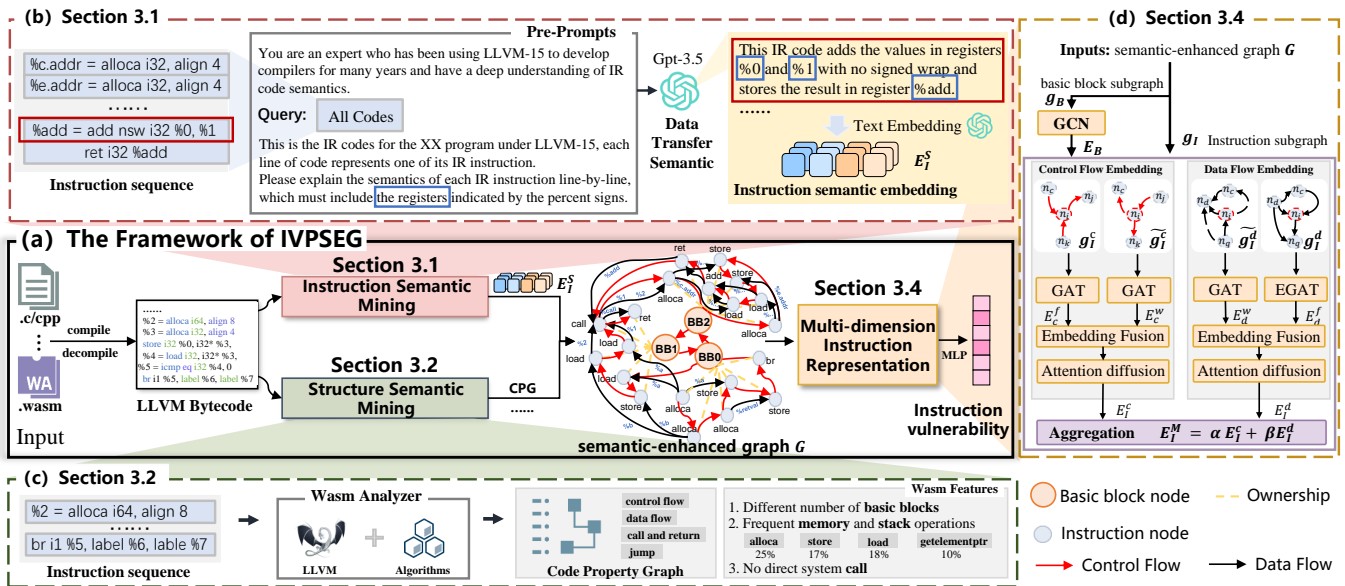

**Figure 2: The overall framework and some details of IVPSEG.**

such as bit flips, and assisting engineers in better discovering Wasm program vulnerabilities before deployment in web applications. We first discuss how to mine the instruction semantics of Wasm programs based on LLM (in Section 3.1). Then, we introduce a program analyzer to effectively extract the contextual structure of Wasm programs (in Section 3.2). Finally, based on the above information, we show how to model error propagation for accurately predicting instruction vulnerabilities (in Sections 3.3 and 3.4).

### 3.1 Instruction Semantic Mining

As discussed above, current methods are limited in the semantic mining of instructions, as they solely rely on human-selected heuristic features to represent instructions. To fully utilize instruction semantics for exploring data transfer in context, we propose a novel framework called instruction semantic mining. Figure 2(b) depicts the overall architecture, which offers an elegant approach for generating instruction embeddings with implicit data transfer. Given a Wasm program $S$, we compile it to an instruction text sequence $\Phi$. Our method initially translates the instruction $n_i$ into a readable semantic text $\vartheta_i$ using prompt expert. Then, we use the pre-trained text embedding model $f_e$ to generate the instruction semantic embedding $E_i^S$.

*3.1.1 Prompt expert.* Our instruction semantic mining begins by configuring a prompt expert to parse the raw instruction $n_i$ while preserving its data transfer semantics. Motivated by the great success of LLM (e.g., ChatGPT[8]) in understanding natural language, we initialize our prompt expert with a specific prompt design using LLM. Specifically, we mine the semantics of raw instructions from the data transfer perspective, as shown below.

**Data Transfer Awareness.** In Web systems, the data for instruction execution is generally transferred through registers (i.e., numerical carriers). In the event of a bit flip, it may propagate with registers between instructions. Therefore, we use LLM to mine the instruction semantics and emphasize the required registers. The primary prompts are shown in Figure 2(b). For instruction text sequence $\Phi$, we prompt LLM to mine the semantics of each instruction

(line-by-line) while preserving the source and destination registers of the instruction. For example, the instruction "%add = add new i32 %0, %1" will be translated as "This IR code adds the values in registers %0 and %1 with no signed wrap and stores the result in register %add."

*3.1.2 Semantic Encoder.* After obtaining the instruction semantic text, we need to mine the inherent data dependencies between instructions. Instead of using shallow embedding models, we aim to use a smaller LLM (Text-embedding-3-small) to encode the semantics of text. In particular, given the semantic text $\vartheta$, the semantic encoder works as follows:

$$E_I^S = f_e(\vartheta), \tag{2}$$

where $E_I^S \in \mathbb{R}^{N \times D}$ denotes the instruction semantic embeddings, and $D$ is the dimension of the embedding vector. Therefore, we can mine the similarity between instructions at the data level to enhance the semantics of data dependency and explore possible error propagation patterns.

### 3.2 Structure Semantic Mining

To further explore the explicit structure semantics of Wasm programs, we build an LLVM-based Wasm analyzer, as shown in Figure 2(c). The key steps are as follows:

*3.2.1 Code Property Graph.* Each IR instruction can be expressed as *{function + instruction syntax}*, where *instruction syntax* consists of opcodes, types, and operands. The standard flow analysis involves obtaining the *control flow* (execution sequence) and *data flow* (data transfer) from each function. To mine implicit error propagation patterns, we extend it to the complete program and construct a code property graph. Besides, instructions with flow relations such as *call* and *jump* are extended to this graph, enhancing the structure semantics of programs, as shown in Appendix Figure 8.

*3.2.2 Wasm Related Features.* It has been shown that the clear correspondence between native and Wasm codes is disrupted due to differences in the number of basic blocks for Wasm and IR[41]. Thus, we split the Wasm program into several basic blocks, which

consist of several instructions, and obtain the dependencies between basic blocks. Besides, the Wasm instructions are stack-based (i.e., operands are stored in a stack) and mainly involve memory operations, so we take the number of memory-related instructions (e.g., *alloca*, *load*), registers used, predecessors, and successors as Wasm related features.

## 3.3 Semantic Enhanced Graph Construction

Up to this point, we have obtained the instruction semantic embeddings $E_I^S$ and code property graph. Now, we will explore how this information can be used for effective program representation.

### 3.3.1 Nodes, Relations, and Features Extraction.
We can extract the basic block nodes $\Delta$ and instruction nodes $\Phi$ from code property graph. Then, according to these nodes, the *control flow* relations between basic blocks, the *control flow* and *data flow* relations between instructions are identified. It is worth mentioning that we use registers, which are numerical carriers in data transfer, as edge features of *data flow* relations to explicitly enhance the structure semantic, based on the semantic text $\vartheta_i$. In addition, we extend the *jump* and the function *call* to these two relations, and the *ownership* between basic blocks and instructions are also represented as a type of relations. Then, based on Wasm properties (section 3.2.2), the basic features of nodes are as follows:

**Instruction Features.** Here, we first take the basic attributes of instructions, such as opcodes ($O_e$), number of operands ($O_d$), and width ($B^t$), as features. Then, considering the spatial structure of Wasm programs, we include the number of predecessors ($P_r$) and successors ($S_r$) as features as well. We also found that different types of instructions have different error rates by analyzing the results of fault injection, as shown in Appendix Figure 9. To this end, we take the type ($T_e$) as one of the features. Finally, we also take the $E_v^S$ as one of the features. In summary, the feature of instruction $i$ can be expressed as a six-tuple $I_i = \{O_e, O_d, B^t, P_r, S_r, T_e, E_v^S\}$.

**Basic block Features.** We take the number of memory related instructions ($N^m$) contained in the basic block, predecessors ($P_d$), and successors ($S_c$) as features. In summary, the feature of basic block $j$ can be expressed as a three-tuple $b_j = \{N^m, P_d, S_c\}$.

### 3.3.2 Graph Construction.
To explore how error propagates, we construct a semantic enhanced code property graph using the extracted nodes, relations, and features, which is a multi-layer heterogeneous graph, as shown in Figure 2(a). The basic blocks are represented by orange nodes, and the instructions are represented by blue nodes. Based on DGL[47], we formally represent the semantic enhanced graph and incorporate features into the attributes of corresponding nodes and registers into the edge attributes of data flow relations.

## 3.4 Multi-dimension Instruction Representation

Here, based on the semantic enhanced graph, we develop an instruction representation model for modeling error propagation patterns. The framework is shown in Figure 2(d) with two main parts: **1)** Since different basic block architectures significantly affect error propagation[14], we use graph convolutional network (GCN)[22] to mine the spatial dependencies of basic blocks. Thus, abnormal jumps can be detected based on unusual contextual relations, and the basic block containing faulty instructions can be identified. **2)**

Then, we divide the instruction graph into control flow and data flow subgraphs and use bi-directional graph attention to mine the effects of execution sequence and data transfer on error propagation, respectively. Let $G = (V, E)$ be an instance of the semantic enhanced graph, $V$ represents the set of nodes, including node features, and $E$ represents the set of edges.

### 3.4.1 Context-dependent Extraction.
From $G$, the basic block subgraph $g_B = (v_B, e_B)$ is extracted, where $v_B \in V$ represents the set of basic blocks, and $e_B \in E$ represents the set of basic block edges. Then, GCN is used to mine context-dependent basic blocks, defined as follows:

$$b_i^{l+1} = \sigma(b^l + \sum_{j \in N(B_i)} \frac{1}{C_{ij}} b_j^l W^l), \quad (3)$$

where $N(B_i)$ represents the neighbor of basic block $B_i$, $C_{ij}$ is the product for the square root of node degree, $l$ represents the number of layers, and $\sigma$ represents activation function. The value $b^0$ of the initial layer is $A_{bb}$. Thus, the basic block embeddings are updated to $B' = \{b_1^l, b_2^l, \ldots, b_M^l\}$. Then, they are transmitted to the instruction layer, where each instruction aggregates the embedding of corresponding basic blocks by tensor splicing. Finally, the raw instruction feature $A_{ins}$ is updated to $I' = \{I_1', I_2', \ldots, I_N'\}$.

### 3.4.2 Error Propagation Pattern Mining.
From $G$, we can extract the instruction subgraph $g_I = (v_I, e_I)$, where $v_I \in V$ represents the set of instructions, and $e_I \in E$ represents the set of instruction edges. To explore the different patterns of error propagation in control flow and data flow separately, we divide $g_I$ into the control flow graph $g_I^c$ and data flow graph $g_I^d$. Then, two different GNNs, i.e., bi-directional graph attention networks, are utilized to extract error propagation patterns for each flow graph.

**Modeling Propagation Patterns in Control Flow.** The control flow is the sequence in which instructions are executed, allowing programs to choose different execution paths based on changes in logic. Indeed, the execution sequence of instructions can be influenced by hardware faults such as bit flips, including conditional branching, function calls, etc. Therefore, given the graph $g_I^c = (v_I, e_I^c)$, where $e_I^c \in e_I$. We use a bi-directional graph attention network to mine the patterns of error propagation in the instruction execution sequence. Specifically, we first use graph attention network (GAT)[6] to capture the error propagation pattern and update the weight of edges in $g_I^c$. The calculation process can be summarized as follows:

$$\alpha_{ij} = softmax(a^T LeakyReLU(W[h_i \| h_j])), \quad (4)$$

$$h_i^{l+1} = \sum_{j \in N(n_i)} \alpha_{ij} W^l h_j^l, \quad (5)$$

where $N(n_i)$ represents the neighbor of instruction $n_i$. We assume that every instruction $n_i$ has an initial representation $I_i'$. Then, we compute the weighted average of the transformed features for neighbor nodes as the new representation of instruction $n_i$. The representation of all instructions can be denoted as $E_c^f$. Besides, to enhance the correlation between nodes from the opposite direction, we construct the reverse graph $\widetilde{g_I^c}$ from $g_I^c$. Then, taking the initial representation $I_i'$ as input, we use GAT to obtain the reverse

representation of instruction $n_i$[45]:

$$q_{ij} = softmax(LeakyReLU(a^T[Wh_i \parallel Wh_j])), \quad (6)$$

$$h_i^{l+1} = \sum_{j \in N(n_i)} q_{ij}^l W^l h_j^l, \quad (7)$$

Thus, we can get the reverse instruction representation $E_c^w$. The updated instruction representation can be calculated by:

$$E_c = W_1^c E_c^f + W_2^c E_c^w, \quad (8)$$

where $W_1^c, W_2^c \in \mathbb{R}^N$ are learnable parameters. And the model can focus on more important instructions and excludes unnecessary features.

**Modeling Propagation Patterns in Data Flow.** The data flow is used to describe the data dependencies between instructions, where data is transferred by registers. Hardware faults, such as data corruption due to bit flips, are more likely to propagate along the data flow. Thus, given the graph $g_I^d = (v_I, e_I^d)$, where $e_I^d \in e_I$. We first use EdgeGAT[30] to capture the error propagation pattern and update the weight of edges in $g_I^d$. The representation update for instruction $n_i$ is given by:

$$h_i' = W_s V_i + \sum_{j \in N(n_i)} \alpha_{ij}(W_n h_j + W_e e_{ij}) \quad (9)$$

where $W_s, W_n, W_e$ are used to denote the learnable weight matrices for instruction features, neighboring instructions, and edge features. The attention weights are obtained by:

$$\alpha_{ij} = softmax(LeakyReLU(a^T[W_n h_i \parallel W_n h_j \parallel W_e e_{ij}])), \quad (10)$$

Thus, we can mine the importance of registers for error propagation, and the representation of all instructions can be denoted as $E_d^f$.

Similarly, we also construct the reverse graph $\widetilde{g_I^d}$ from $g_I^d$, and can get the reverse instruction representation $E_d^w$ by equation(6-7). Finally, the updated instruction representation can be calculated by:

$$E_d = W_1^d E_d^f + W_2^d E_d^w, \quad (11)$$

where $W_1^d, W_2^d \in \mathbb{R}^N$ are learnable parameters.

**Modeling Multi-hop Propagation Patterns.** In fault injection experiments, we have found that hardware faults propagated along instruction execution and data transfer over multi-hop. Thus, we only utilize one layer to update instruction representation, but introduce multi-hop neighbors in the single-layer message propagation, which can contribute more patterns of the error propagation and reduce the over-smoothing problem. Inspired by related work[25, 46], we define the multi-hop attention diffusion layer as:

$$H^{k+1} = W_\alpha^k A H^k + W_\beta^k H, \quad (12)$$

where $A$ is the one-hop attention matrix, $k$ is the number of hops, $H$ is the initial input, and $W_\alpha, W_\beta \in \mathbb{R}^k$ are learnable parameters, $W_\alpha^k + W_\beta^k = 1$. With $E_c$ and $E_d$ as inputs, respectively, we use this mechanism to obtain the instruction embedding $E_I^c$ and $E_I^d$. This not only expands the receptive field of the target instruction but also adapts to changes in the execution of instructions. Finally, we perform a weighted summation of $E_I^c$ and $E_I^d$ to measure the impact of control flow and data flow on instruction vulnerability prediction.

Then, through a linear layer, we can obtain the predicted instruction vulnerabilities $Y$:

$$E_I^M = LeakyReLU(\alpha E_I^c + \beta E_I^d), Y = softmax(MLP(E_I^M)), \quad (13)$$

where $\alpha, \beta$ are the training parameters. And the cross-entropy loss is defined as:

$$\mathcal{L} = -\sum_{i=1}^c y \ln Y + \lambda \sum_{i=1}^p |\theta_i|, \quad (14)$$

where $p$ is the total parameter of our model.

## 4 EXPERIMENTS

In this section, we perform comprehensive experiments to validate the effectiveness of our method. We aim to address the following research questions:

**RQ1** (See §4.3): What is the performance of IVPSEG compared to state-of-the-art methods regarding prediction accuracy, prediction quality[4] on vulnerable instructions and model robustness against different training sample sizes?

**RQ2** (See §4.4): What is the effect of each module in IVPSEG? For C1 (insufficient structure semantic) and C2 (lack of instruction semantic), is the performance improvement attributed to the semantic enhanced graph and GNNs we propose?

**RQ3** (See §4.5): Is our method effective in error propagation modeling, and how does IVPSEG perform in real-world Wasm programs?

### 4.1 Implementation

*4.1.1 Wasm Compilation and Fault Generation System.* The system is deployed on a high-performance machine equipped with an Intel(R) Core(TM) i7-14700KF CPU, 64 GB of running memory, and the operating system Ubuntu 20.04.

**Wasm Compilation:** As the first stage of Wasm fault injection, Wasm compilation is used to translate Wasm programs to IR. Specifically, given a Wasm program, we first translate it to native code using WABT-based *wat2wasm* and *wasm2c*, which includes a library that simulates Wasm memory and stack operations, and a mapping of functions and data structures. Then, we use *clang* to compile the native code to LLVM IR, which serves as the input for the Wasm fault injection model.

**Wasm Fault Injection Model:** At present, the knowledge about the impact of bit flips on Wasm programs is scarce, so we design an autonomous hardware fault injection tool based on LLFI. The main parameters of the fault model $\Psi^M$ are $\{S_e, F_T, N^F, Reg\}$, where $S_e$ is the instruction type for fault injection (i.e., all instructions), $F_T$ is the type of fault (i.e., bit flip), $N^F$ represents the total cycle of fault injections (i.e., max 10000), and $Reg$ is the registers for fault injection (i.e., *desreg*, *srereg1*), which are numerical carriers in data transfer. Based on the $S_e$ and $Reg$, we get the register bit-width $R_n^b$ of the corresponding instruction, and then flip its machine code bit by bit, to realize the effect of random bit flips during the execution of the program. Finally, the error rate for each instruction is calculated as follows:

$$P_I = \frac{Num_{err}}{R_n^b \times N^F}, \quad (15)$$

---

[4]It is a measure of how close the model's predictions are to the actual values.

Table 1: Comparison of results for instruction vulnerability prediction in benchmarks.

| Program | GATPS | | PrograML | | DegraphCS | | PerfoGraph | | MPIGNN | | IVPSEG | |
|---|---|---|---|---|---|---|---|---|---|---|---|---|
| | Acc | F1 | Acc | F1 | Acc | F1 | Acc | F1 | Acc | F1 | Acc | F1 |
| Basicmath | 88.0±0.5 | 92.7±0.3 | 89.7±1.0 | 93.7±0.6 | 88.5±0.8 | 93.0±0.5 | 85.7±0.8 | 91.1±0.5 | 87.4±1.0 | 92.3±0.6 | **90.3±1.0** | **94.2±0.6** |
| Dijkstra | 72.8±0.9 | 25.2±3.2 | 81.8±1.3 | 42.9±4.5 | 84.0±0.7 | 50.5±3.7 | 83.6±1.6 | 57.9±4.9 | 81.4±0.9 | 31.6±7.8 | **90.0±0.6** | **75.0±1.3** |
| Qsort | 66.8±2.6 | 70.1±2.8 | 65.2±3.0 | 67.0±2.4 | 63.6±2.4 | 65.4±1.6 | 59.4±1.4 | 60.2±1.1 | 62.4±0.9 | 68.1±0.7 | **77.4±1.7** | **77.5±2.1** |
| Isqrt | 95.0±1.1 | 96.9±0.6 | 93.7±1.8 | 96.0±1.1 | 95.0±0.0 | 96.7±0.7 | 97.5±0.0 | 98.3±0.8 | 95.0±1.1 | 96.7±0.7 | **99.9±0.0** | **99.9±0.0** |
| Float-mm | 74.4±1.1 | 70.1±2.4 | 60.0±2.1 | 55.6±4.5 | 59.3±2.9 | 57.5±2.8 | 74.4±3.5 | 67.1±4.3 | 60.0±1.5 | 47.1±4.4 | **88.2±0.7** | **86.1±1.0** |
| Fft | 75.1±2.1 | 79.5±1.6 | 72.0±1.3 | 78.2±1.3 | 68.8±1.0 | 75.0±1.2 | 77.3±1.1 | 81.3±1.0 | 70.6±0.7 | 77.3±0.5 | **87.5±1.0** | **89.2±0.8** |
| N-body | 77.0±0.8 | 84.9±0.6 | 77.5±0.4 | 85.7±0.3 | 78.2±0.5 | 86.3±0.3 | 77.5±1.5 | 85.0±1.1 | 76.4±0.4 | 84.9±0.3 | **82.5±1.5** | **87.5±1.2** |
| Towers | 71.8±2.2 | 64.1±4.0 | 77.2±3.3 | 65.5±3.9 | 75.1±1.6 | 71.0±2.4 | 77.2±1.8 | 71.7±1.6 | 70.8±3.2 | 63.0±4.5 | **82.7±0.9** | **78.2±1.5** |
| Factorial | 72.9±2.1 | 71.3±2.5 | 68.2±2.6 | 62.4±3.5 | 70.5±1.6 | 70.2±2.3 | 65.8±1.9 | 63.5±2.7 | 78.8±2.1 | 80.0±1.7 | **90.5±2.1** | **89.6±2.3** |
| Rot | 75.4±0.8 | 77.5±0.7 | 80.0±1.1 | 72.7±1.7 | 69.1±0.8 | 69.8±0.8 | 75.0±0.5 | 74.0±1.0 | 69.5±0.5 | 71.1±0.5 | **85.2±1.5** | **84.9±1.4** |

where $Num_{err}$ is the number of errors that occurred in the program. Based on this model, the bit flips are applied to partial instructions of Wasm programs, resulting in a total of 1,070,000 fault samples.

*4.1.2 Instruction Vulnerability Prediction Model.* Based on the IR and fault samples obtained by the above system, we construct an instruction vulnerability prediction model. Specifically, the model is implemented in Pytorch-1.10.2 with Adam optimizer. The learning rate is set to 0.005. *LEAKY_RELU* is applied as the activation function. GPT-3.5 and Text-Embedding-3-small are used for semantic mining and representation, respectively. The dimension of instruction semantic embeddings is set to 128. We divide the IR instructions into two sets: 80% for training, and the remaining instructions for testing. We select the one with the best performance in the validation set and then evaluate it on the test set. All hyperparameters are tuned based on the performance of the validation set.

## 4.2 Experiment Setup

*4.2.1 Dataset.* Following previous studies[7, 16, 48], we conduct experiments on common benchmarks (i.e., MiBench[17] and Jet-Stream2[18]). We select the most representative programs from these benchmarks, as they are widely employed in Wasm evaluations and relevant studies[28, 35, 44]. A concise overview of the programs employed in our experiments is provided in Appendix Table 4, including Basicmath, Dijkstra, Qsort, Isqrt, Float-mm, Fft, N-body, Towers, Factorial, and Rot. For these programs, we utilize the above system to obtain the IR instructions and fault samples.

*4.2.2 Baselines.* In our experiments, we compare our method with five state-of-the-art methods.

- **GATPS**[29], which uses the program relation graph and the encoding of instructions to predict instruction vulnerabilities.
- **PrograML**[11], which constructs a graph representation of the program based on IR and adapts gated graph neural networks to extract node embeddings.
- **DegraphCS**[54], which uses variable-based flow graphs to represent programs and utilizes an improved gated graph neural network with an attention mechanism to learn instruction representation.
- **PerfoGraph**[43], which captures numerical information and data structure by introducing new nodes and edges, and proposes an adapted embedding method to incorporate data awareness.

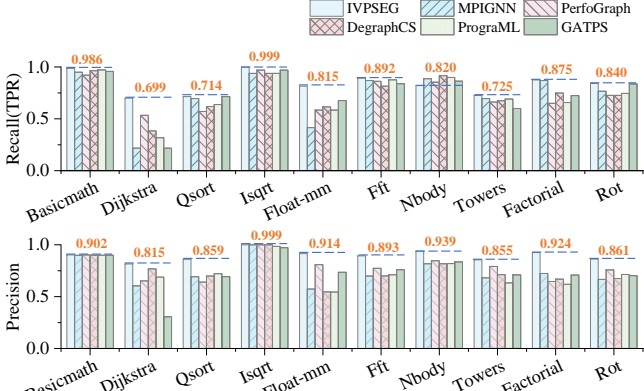

Figure 3: Performance of different methods in prediction quality for vulnerable instructions.

- **MPIGNN**[13], which utilizes embeddings and graph attention convolution to tackle the issue of identifying errors in programs.

## 4.3 Overall Results

To answer **RQ1**, we conduct extensive experiments on benchmark programs for instruction vulnerability prediction. The experimental results are comprehensively evaluated by four metrics: accuracy (*Acc*), precision (*Pre*), recall (*TPR*), and *F*1-score. The performance of different training sample sizes is also evaluated. Table 1, Figure 3, and Figure 4 present the results of IVPSEG compared to other baselines. We can make the following observations.

① **Our method significantly outperforms the state-of-the-art methods in all programs.** In Table 1, IVPSEG consistently outperforms all baselines across 10 Wasm programs. Specifically, compared to the most competitive baseline, our method improves 0.6%-18.5% in *Acc* and 0.5%-22.8% in *F*1. Additionally, our method exhibits excellent adaptability, achieving up to 77.4% accuracy even in the worst-performing Qsort program. This superiority can be attributed to the advantage of the proposed semantic enhanced graph and GNNs, which augments the instruction representation from the inherent semantic and structure semantic. Thus, IVPSEG can be used to analyze the instruction vulnerabilities of Wasm programs during the stages of Web development and testing.

② **IVPSEG's prediction quality for vulnerable instructions is superior to most baselines.** Based on the prediction values and the error rates obtained by fault injection, the *Pre* and *TPR* are

Table 2: Ablation performance of different variants.

| Model | Acc | Pre | F1 |
|---|---|---|---|
| IVPSEG-ns | 86.7±1.7 | 88.9±1.7 | 85.4±1.6 |
| IVPSEG-nr | 85.6±1.0 | 80.0±3.1 | 79.6±1.1 |
| IVPSEG-nc | 85.7±1.2 | 87.8±3.2 | 82.3±2.4 |
| IVPSEG-nd | 84.8±1.7 | 84.6±0.9 | 84.5±0.9 |
| **IVPSEG** | **87.7±1.0** | **90.6±1.6** | **87.0±1.3** |

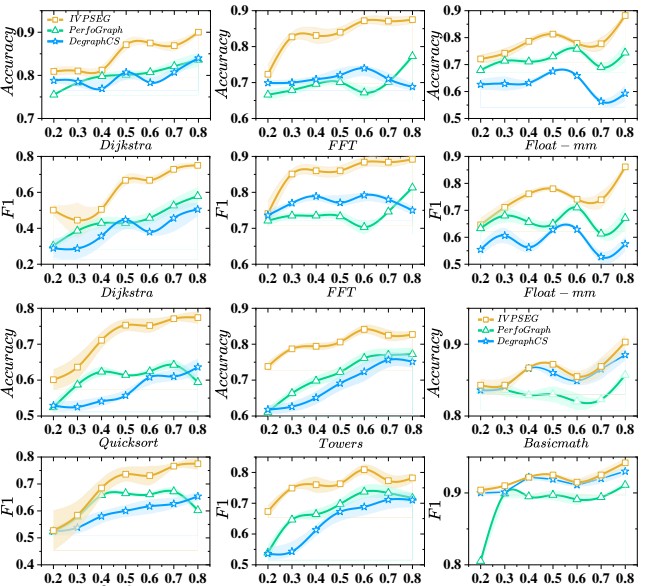

Figure 4: Performance at different training sample sizes. From the fault samples, 20~80% of instructions are randomly selected to train the model, and then the remaining instructions are used to evaluate the prediction performance.

derived for vulnerable instructions, and the results are shown in Figure 3. It can be seen that our method has a good performance for truly vulnerable instructions. Specifically, compared to the most competitive baseline, our method improves 6%-27.8% in *Pre* and 0.6%-31.1% in *TPR*. Although on the N-body program, IVPSEG has only 82% in *TPR*, its *Pre* is as high as 93.9%. It suggests that our method can better predict truly vulnerable instructions.

③ **Our method has better robustness at different training sample sizes.** To explore how many fault instructions IVPSEG needs to achieve robust performance, we randomly take a certain amount of instructions (20~80%) from the training set to retrain the model and evaluate the accuracy and F1-score, as shown in Figure 4 and Figure 10 (Appendix). It can be seen that our method always outperforms the baselines, even at small training samples, which demonstrates that IVPSEG can better derive instruction vulnerabilities from contextual semantics. Besides, the effect of IVPSEG at small fault samples is similar to that of baselines at large fault samples (e.g., in the Fft program, the accuracy of IVPSEG is 0.827 at 30% fault instructions, while PerfoGraph's accuracy is 0.773 at 80% fault instructions). Thus, our method has high training efficiency, and its performance is better even with a small number of fault instructions.

### 4.4 Ablation Study

For **RQ2**, several variants of IVPSEG are introduced as other comparisons:

- **IVPSEG-ns**, which removes the instruction semantic mined by LLM;
- **IVPSEG-nr**, which removes the reverse graph attention;
- **IVPSEG-nc**, which removes the control flow of instructions;
- **IVPSEG-nd**, which removes the data flow of instructions.

For each ablation, we train the model from scratch using an equivalent experimental setup while varying individual components. The results are shown in Table 2.

① **The effect of the semantic enhanced graph.** It can be seen that the predicted effect of IVPSEG-nd is significantly reduced. The *Pre* decreased by 6%, *Acc* and *F*1 decreased by 2.9% and 2.5%, respectively, since the data flow is highly dependent on memory and registers. When bit flips occur in registers or memory, these errors may be loaded into specific instructions and propagate with data transfer, affecting the execution of Wasm programs. Additionally, the semantics of instructions (IVPSEG-ns) and control flow (IVPSEG-nc) also have an impact on instruction vulnerability prediction (decreased by 1%-4.7%). It suggests that our semantic enhanced graph can represent Wasm programs well and explore error propagation patterns.

② **The effect of the bi-directional graph attention.** From Table 2, we observe a noticeable performance decline when we only keep the normal attention (IVPSEG-nr). Specifically, The *Pre* decreased by 10.6%, *Acc* and *F*1 decreased by 2.1% and 7.4%, respectively. This indicates that the introduction of reverse graphs augments the dependencies between instructions, which provides an improved way to mine error propagation patterns.

### 4.5 Case Study

*4.5.1 Error Propagation Analysis.* For **RQ3**, we first utilize visualization to analyze how error propagates by examining the learned edge weights of IVPSEG. In Figure 5, we present the representation of edge weights learned by IVPSEG for the Factorial program. Darker colors indicate greater weight values, suggesting a stronger influence on adjacent instructions and a higher probability of error propagation with the edge. From Figure 5, it is evident that IVPSEG mines potential error propagation patterns well. For example, our method recognizes that the No.17 instruction largely propagates the error along the No.49, to No.50 instruction, rather than along the No.20-24 instruction. In fact, with LLFI, we find that the error result may be returned through the corresponding registers "%var → %12" in the event of No.17 errors. Additionally, since the No.24 instruction overwrites the error value, the path "%var → %1 → %2 → %3 → %conv" has little effect on the program. **Thus, the essential propagation path of the error in the data flow can also be more precisely determined by IVPSEG.** It is important to note that data is not typically accessed in the exact order of program execution, but only the instructions utilizing the data receive it. As a result, errors are often propagated backward with the execution of the data flow.

*4.5.2 Performance on Real-world Wasm Programs.* Then, we choose one of the most popular Wasm benchmarks from GitHub, called *wasm32-wasi-benchmark*[5], and perform instruction vulnerability

---

[5] https://github.com/second-state/wasm32-wasi-benchmark

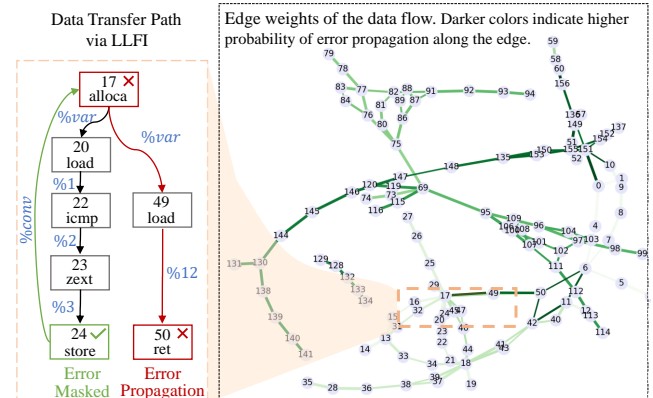

**Figure 5: The edge weights of the data flow learned by IVPSEG on the Factorial program. The left box is the real data transfer path captured by LLFI.**

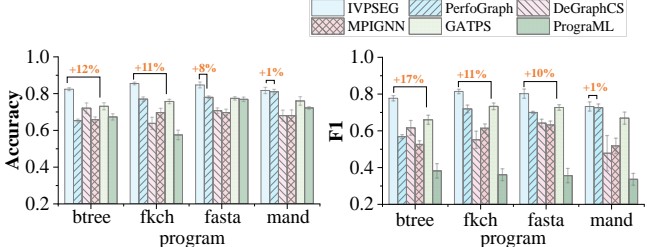

**Figure 6: Comparison of Accuracy and F1 for instruction vulnerability prediction in real-world Wasm programs.**

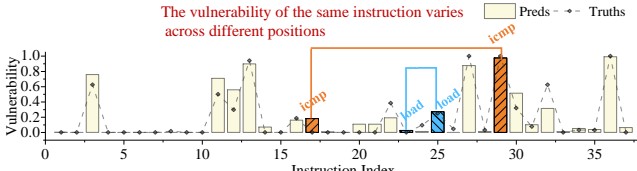

**Figure 7: Predicted instruction vulnerabilities vs ground truths from fault injection in the fkch program.**

prediction for all of these available Wasm programs, the results are shown in Figure 6 and Figure 11 (Appendix). As can be seen, compared to the most competitive baselines, IVPSEG improves performance by 1~12% in *Acc* and 1~17% in *F*1, which suggests that our method can well help developers to understand the error resilience of Wasm programs before deployment. Then, corresponding measures can be adopted to improve the security of Web systems.

Additionally, we notice an interesting phenomenon: **The vulnerability of the same instruction varies across different positions.** As depicted in Figure 7, the vulnerability of the No.29 *icmp* instruction is as high as 0.98, while the No.17 is only 0.01. The No.17 instruction is *"%cmp31 = icmp sgt i32 %33, 0"*, after the bit flip occurs, the probability that %33's value less than 0 is very low, so it will not affect the subsequent instruction to run. And the No.31 instruction is *"%cmp142 = icmp sgt i32 %99, %100"*, its vulnerability is highly dependent on "%99" and "%100". Our method demonstrates superior accuracy in predicting instruction vulnerabilities across different semantics, enabling efficient identification of high-vulnerability positions in the program where redundancy can be implemented to minimize costs.

## 5 RELATED WORK

This section summarizes the existing literature related to this work, which includes Wasm program graph representation and instruction vulnerability prediction.

### 5.1 Wasm Program Graph Representation

Due to the syntax and semantic structure of the program, it is natural to represent it as graph[3], which can be utilized for learning semantic embeddings[49] and detecting program vulnerability[10]. For example, Cabrera-Arteaga et al.[9] leveraged an e-graph data structure to represent the Wasm program by analyzing its expressions and operations through the data flow. Breitfelder et al.[5] developed a static analysis framework for Wasm, which can provide some necessary information for vulnerability detection, such as control flow and data flow. TehraniJamsaz, et al.[43] proposed a graph-based program representation, which aggregated data types and provided numerical awareness, making it highly effective for performance optimization tasks. Despite the availability of some graph representations of programs, they were not well adapted to instruction vulnerability prediction, lacking hierarchical structure and inherent semantics of instructions.

### 5.2 Instruction Vulnerability Prediction

Currently, the field had two main categories: 1) Vulnerability prediction based on fault injection[20, 39]. These methods generated errors by simulating hardware faults and identified vulnerable instructions through statistical analysis. For example, Agarwal et al.[1] proposed a framework-agnostic fault injection tool for programs, allowing users to run fault injection at the IR level and better understand how faults propagate between instructions. Sharma et al.[40] employed coverage-guided software fault injection to detect application errors, which was generic and targeted to explore a given program's error handling behavior effectively. However, the cost of hardware fault injection increases with program size. 2) Vulnerability prediction based on artificial intelligence[26, 34]. These methods built a dataset by performing partial fault injection on program instructions to train the model and identify error-prone instructions. For example, by creating a heterogeneous graph of program instructions and utilizing a graph attention network, Ma et al. [29] proposed a graph attention network, which was able to predict the different sorts of errors.

## 6 CONCLUSIONS

In this paper, we proposed a novel paradigm, IVPSEG, which could accurately predict instruction vulnerabilities and was applicable to a variety of Wasm programs. Specifically, we first used GPT to automatically extract semantic embeddings, which contain the semantic knowledge of instructions in context. Then, we utilized semantic embeddings and program structure to construct a semantic enhanced graph, which implicates the potential path of error propagation. Based on this graph, we designed graph neural networks and attention diffusion to predict instruction vulnerabilities by modeling the spatial dependency between instructions and capturing different error propagation patterns. Finally, we built a Wasm compilation and fault generation system, where we can simulate register or memory bit flips, which are numerical carriers for data transfer. The experimental results on the Wasm benchmarks demonstrated the effectiveness of our method.

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

# APPENDIX

## A Variable Notations

We list the used variable notations in Table.3.

**Table 3: Frequently used notations.**

| Notations | Descriptions |
|---|---|
| $\Phi$ | The LLVM IR instruction sequence. |
| $n_i$ | An instruction, $i = 1, 2, ..., N$. |
| $N$ | Total number of instructions. |
| $\Delta$ | The basic block sequence. |
| $B_i$ | A basic block, $i = 1, 2, ..., M$. |
| $M$ | Total number of basic blocks. |
| $J$ | The number of relations; |
| $\Lambda$ | The set of all features. |
| $E_i^S$ | The semantic embedding of instruction $i$. |
| $I_i$ | The feature of $n_i$. |
| $b_j$ | The feature of $B_j$. |
| $G$ | A semantic enhanced graph with $V$ and $E$. |
| $V$ | The set of nodes in $G$. |
| $E$ | The set of edges in $G$. |
| $g_B$ | A subgraph of basic block $g_B \in G$. |
| $v_B$ | The set of instructions in $g_B$, $v_B \in V$. |
| $e_B$ | The set of edges in $g_B$, $e_B \in E$. |
| $B'$ | The set of updated basic block embedding. |
| $g_I$ | A subgraph of instruction $g_I \in G$. |
| $v_I$ | The set of instructions in $g_I$, $v_I \in V$. |
| $e_I$ | The set of edges in $g_I$, $e_I \in E$. |
| $g_I^c$ | The control flow subgraph, $g_I^c \in g_I$. |
| $g_I^d$ | The data flow subgraph, $g_I^d \in g_I$. |
| $E_c$ | The updated instruction embeddings in $g_I^c$. |
| $E_d$ | The updated instruction embeddings in $g_I^d$. |
| $E_I^M$ | The set of instruction embeddings. |
| $Y$ | The set of predicted instruction vulnerabilities. |

## B Code Property Graph

Here, we show the extracted graph using the Add program for example, as shown in Figure 8.

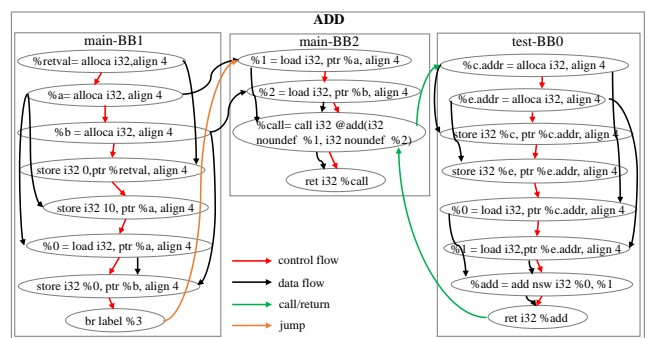

**Figure 8: The code property graph of the Add program.**

## C Error Rate Statistics

We divide instructions into 8 types based on the official LLVM standard. We carry out fault injection to programs in the benchmark based on the system defined in this paper, and calculate the error rate for each type of instruction, and finally sum-average the result of each program. The concise overview and results are shown in Figure 9. The "mem-op" denotes the operations on system memory, such as $alloca, store$. The "ter-op" denotes the termination of basic blocks or functions in programs, such as $br, ret$. The "cast-op" denotes the type-forced conversion, such as $bitcast, sext$. The "comp-op" denotes the data used for comparison, such as $icmp, fcmp$. The "int-op" denotes the integer binary operation, such as $sub, div$. The "float-op" denotes the floating-point binary operation, such as $fmul, frem$. The "logic-op" denotes the logical or shift operation, such as $lshr, and$. The "other-op" denotes other types of instructions, such as $phi, select$.

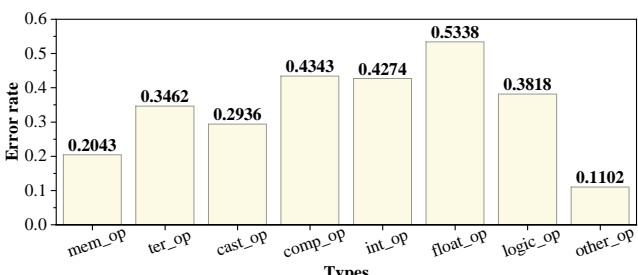

**Figure 9: Error rates for different types of instructions.**

## D Overview of Wasm Programs

A concise overview of programs employed in our experiments is provided in Table 4.

**Table 4: Statistics of programs studied in our experiments. Float-mm (floating point matrix multiplication), Fft (fast fourier transform), N-body (multibody problem), Towers(tower of hanoi), and Rot (encryption and decryption). These programs consist of hundreds of code segments, each configured with a test suite.**

| Programs | Instructions | Control and Data flow | Faults Injected |
|---|---|---|---|
| Basicmath | 201 | 215 + 186 | 101029 |
| Dijkstra | 319 | 343 + 271 | 142030 |
| Qsort | 211 | 231 + 210 | 154147 |
| Isqrt | 87 | 92 + 90 | 66107 |
| Float-mm | 167 | 180 + 155 | 107249 |
| Fft | 252 | 261 + 254 | 114059 |
| Nbody | 440 | 446 + 497 | 131412 |
| Towers | 267 | 299 + 253 | 148695 |
| Factorial | 162 | 175 + 160 | 17340 |
| Rot | 547 | 589 + 576 | 96358 |

## E Robustness at Different Training Sample Sizes

To explore how many fault instructions IVPSEG needs to achieve robust performance, we randomly take a certain amount of instructions (20 ~ 80%) from the training set to retrain the model and evaluate the accuracy and F1-score. The results of other programs are shown in Figure 10.

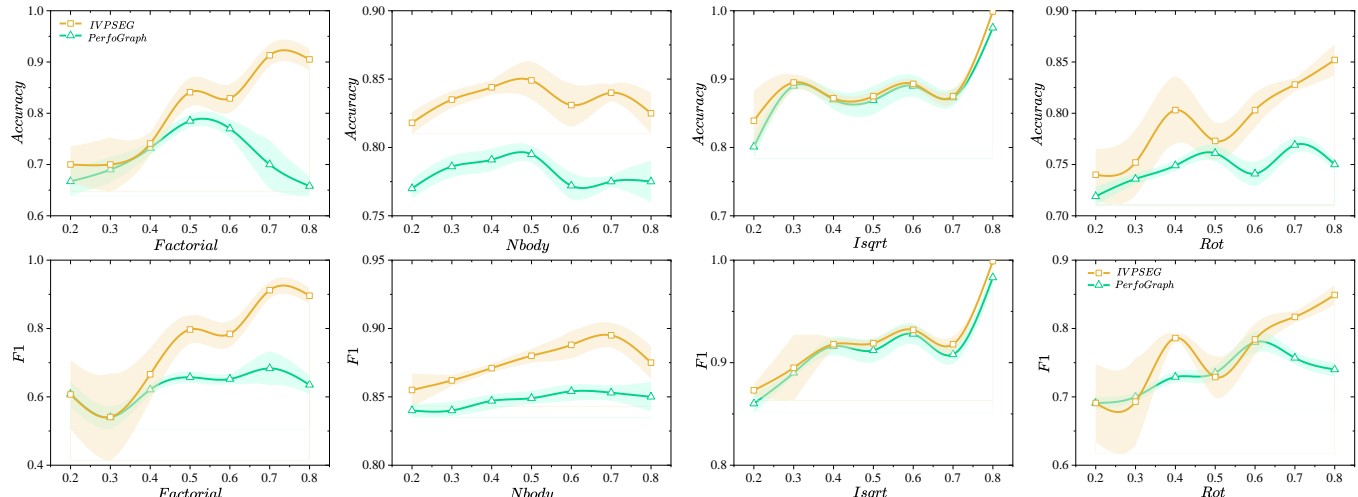

Figure 10: Performance at different training sample sizes on other Wasm programs

## F  Performance on Real-world Wasm Programs

Based on the prediction values and the error rates obtained by fault injection, the *Pre* and *TPR* are derived for vulnerable instructions, and the results are shown in Figure 11. As can be seen, compared to the most competitive baselines, IVPSEG improves performance by 6∼19% in *Pre* and 1∼16% in *TPR*, which suggests that our method can better predict truly vulnerable instructions.

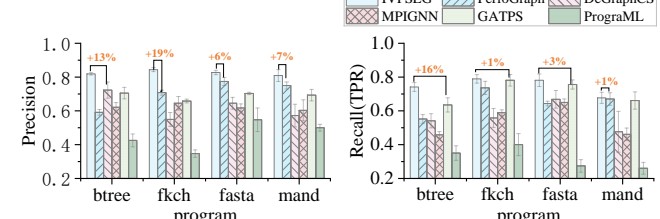

Figure 11: Comparison of Precision and Recall (TPR) for instruction vulnerability prediction in real-world Wasm programs

