# OpenReview forum: "Instruction Vulnerability Prediction for WebAssembly with Semantic Enhanced Code Property Graph"
_ACM.org/TheWebConf/2025/Conference — WWW 2025 Poster_

### Official Review · Reviewer_GtLN · 2024-11-27

**Novelty:** 7
**Technical Quality:** 5

**Review:**

### Summary
The paper proposes IVPSEG, a novel system to capture
error propagation within WebAssembly (wasm) programs.
IVPSEG adopts a Large Language Model to mine
the semantics of instructions. It also uses graph neural networks
and attention diffusion to optimize instruction embeddings
for capturing error propagation patterns.
The experiments demonstrate that IVPSEG is superior to
existing five approaches (e.g., GATPS, PrograML,
DegraphCS, PerfoGraph, MPIGNN) in performance.

### Strengths
[+] Modeling to mine the contextual semantics of instructions,
including structures (data/control flow) and as well as instructions
[+] Building a prototype for the fault generation and prediction framework
[+] Empirical results that demonstrate performance improvement
over five prior baselines

### Weaknesses
[-] Unclear threat model
[-] Missing discussions and limitations
[-] Missing the evaluation of efficiency
[-] Presentation errors

### Comments
Thanks for submitting the paper to WWW '25.
This paper is an interesting read and overall well-written.
However, I have several concerns as follows.

First, the threat model needs to be described.
Is the proposed technique to mitigate the side-channel
attacks like RowHammer? I wonder if there is another assumption
or case that causes bit flips. Please clarify.

Second, equations are difficult to follow as
some notations are missing, e.g., a, h_{i}, h_{j} in Eq(4).
Similarly, how is C_{ij} computed in Eq(3)?

Third, I suggest the authors include discussions and limitations
so that one can draw boundaries on what the proposed
approach can and cannot do. For example, the authors
train a static model with a limited dataset. Would the model
be efficient for other programs in general?

Fourth, the paper does not include the evaluation of IVPSEG
efficiency. I wonder how fast the analysis can be performed
given the complex design of the proposed approach.

Fifth, the paper's presentation needs to be enhanced.

### Miscellaneous
- In Sec. 1, the resource consumption grows exponentially with program size
  -> their resource consumption grows exponentially with the size of the program.
- In Fig. 1, error data -> data error; error address -> address error
- In Fig. 2, (d) should point to Section 3.3
- In Table 1, the citations for baselines require to be inserted
- In Fig. 2(a), Sec. 3.3 is missing
- In Sec. 3.3 six-tuple -> seven-tuple
- In Tbl 1, inserting an average of whole programs at the last row would be helpful

**Questions:**

- In Def. 1, is it a common definition of instruction vulnerability in the literature
when a program incorrectly runs with the instruction changes due to bit flips?
To me, this sounds misleading as it looks like an instruction itself is vulnerable.
Moreover, the definition that a vulnerability is a probability sounds ambiguous.
- In Sec. 3, the authors state that the IVPSEG framework can assist in discovering
those faults *before* web application deployment. How likely are such hardware
faults (e.g., bit flips) to occur, assuming there are no external attacks?
- In Sec. 3.1, "Fig. 2(b) depicts the overall architecture" ->
Isn't it Fig. 2(a) that depicts the whole framework?
- What is the purpose of getting reverse instruction representation?
- "𝑝 is the total parameter of our model" -> What does this mean?
- In Tbl 1, what does underlined accuracy / F1 represent?
- How does IVPSEG handle indirect calls?

**Reviewer Confidence:**

2: The reviewer is willing to defend the evaluation, but it is likely that the reviewer did not understand parts of the paper

**Scope:**

4: The work is relevant to the Web and to the track, and is of broad interest to the community

---

### Official Review · Reviewer_ZEhd · 2024-11-28

**Novelty:** 5
**Technical Quality:** 3

**Review:**

Dear authors, thanks for submitting your work to The Web Conf'2025. This paper proposes a set of interesting techniques for vulnerability detection under bit flip induced failures. I do like how you integrated LLVM program analysis with LLM and GNN, and I do believe you have some insights on instruction level vulnerabilities.  That being said, I think this paper needs significant clarification/improvement on the following aspects:

1. Motivation. The authors motivate this paper by claiming bit flip incidents as an attack vector to Wasm applications. However, the paper does not show any evidence that such claim holds. The author explicitly mentioned in 4.1 Implementation that "at present, the knowledge about the impact of bit flips on Wasm programs is scarce." If this is the case, then it seems the paper should start with investigations (e.g. user studies, monitoring etc.) the impact of bit flips on Wasm. Without concrete evidence that such problem actually exist, it is hard for readers/reviewers to judge whether this paper is working on an important problem. Being the first to study a problem is definitely a good thing, but it also means you have the responsibility to prove its importance and relevance to the community. The concrete suggestion would be to explicitly clarify how frequent is bit flip incidents in Wasm systems, and what are the the consequences of such incidents.

2. Technical correctness. Overall IVPSEG (the name is very hard to remember BTW) proposes a set interesting techniques, but my concerns are as follows: 1) Why is the proposed techniques specific to Wasm? It seems most techniques can be applied to a broader array of frameworks, e.g. eBPF etc., so why does this paper limits itself to Wasm? What are some domain specific challenges that IVPSEG address? What would happen if we use IVPSEG for other software frameworks? 2) Why is ML needed for vulnerability detection? Why is LLVM IR/Wasm binaries the correct layer to work on? Suppose there are actual bit flips in the Wasm binaries before they are deployed/instantiated, then why not apply simple pre-flight checks to compare Wasm binaries with the ground truth (e.g. diff local code with upstream to detect any delta). If the bit flips occur after deployment/instantiation, then there is not much to be done because the impact has already been made? In other words, when and why should Wasm DevOps engineers use ML-based detections?

3. Evaluation validity. The evaluation overall makes sense to me, my questions are mainly on how well could IVPSEG generalize to other unseen Wasm programs. It would be great to have not only a tool to inject fault, but also something to gather/synthesize large amount of Wasm applications, so that the paper can evaluate whether the proposed approach is generalizable or not. I don't feel this is necessary to make this paper acceptable, but it would be a good-to-have.

Overall, I like the techniques proposed in the paper, but I have some concerns regarding the motivation and technical roadmap. I'm looking forward to the response from authors.

**Questions:**

The questions are mostly listed in my review, for instance:
1. Is bit flip possible within Wasm ecosystem? What are some concrete evidences? What experiments can be done to prove this?
2. Why is the proposed techniques specific to Wasm? Why does this paper limits itself to Wasm? What are some domain specific challenges that IVPSEG address? What would happen if we use IVPSEG for other software frameworks?
3. Why is ML needed for vulnerability detection? Why is LLVM IR/Wasm binaries the correct layer to work on? Why not apply simple pre-flight checks to compare Wasm binaries with the ground truth (e.g. diff local code with upstream to detect any delta)?
4. How well does IVPSEG generalizes to unseen Wasm applications?

**Reviewer Confidence:**

3: The reviewer is confident but not certain that the evaluation is correct

**Scope:**

3: The work is somewhat relevant to the Web and to the track, and is of narrow interest to a sub-community

---

### Official Review · Reviewer_iSG9 · 2024-11-30

**Novelty:** 6
**Technical Quality:** 6

**Review:**

The paper addresses a critical and emerging issue in WebAssembly (Wasm), bit-flip vulnerabilities which are caused by factors such as RowHammer attacks and voltage scaling. By focusing on this relatively unexplored area, the research provides valuable insights into the security risks associated with Wasm programs, which are gaining popularity in web applications.

The proposed IVPSEG framework is also noteworthy and innovative. Authors have done a thorough evaluation, where they validated IVPSEG on a set of 14 benchmark programs, demonstrating its good performance w.r.t state-of-the-art methods in terms of accuracy and robustness.

Overall, the paper makes impactful contributions to advancing WebAssembly security, offering a sophisticated framework for identifying and addressing Wasm instruction vulnerabilities.

**Questions:**

Although the approach followed by the authors is innovative and well-evaluated, I would like to ask the below queries to understand the authors' perspective:

1. The framework’s heavy reliance on Large Language Models (LLMs) and complex Graph Neural Networks (GNNs) may increase computational costs, potentially making IVPSEG resource-intensive and slower to deploy, especially for larger Wasm programs. Could the authors clarify how the framework addresses this?

2.  In addition to the control and data flow graphs (CFG and DFG), other semantic information graphs (such as AST or PDG) could be relevant. Did the authors consider these for comparison?

**Reviewer Confidence:**

3: The reviewer is confident but not certain that the evaluation is correct

**Scope:**

4: The work is relevant to the Web and to the track, and is of broad interest to the community

---

### Official Review · Reviewer_MGnz · 2024-12-02

**Novelty:** 4
**Technical Quality:** 4

**Review:**

### Summary
This paper studies the vulnerabilities in WebAssembly (Wasm) instructions, particularly focusing on the impact of hardware faults such as bit flips caused by voltage scaling and RowHammer attacks. Existing works have not adequately addressed the specific vulnerabilities of Wasm programs, especially considering their unique features compared to traditional programs. To address this limitation, the author proposes using a semantic enhancement to the Code Property Graph (CPG) to predict and mitigate these vulnerabilities. The experimental results on various datasets demonstrate the effectiveness of this approach in identifying potential vulnerabilities in Wasm code.

### Strength
* The paper is well written.
* The topic is interesting and practical.
* The source code is public.

### Weakness
* There is a related work, "Discovering Vulnerabilities in WebAssembly with Code Property Graphs", which is failed to be discussed in this paper.
* In this paper, LLMs and GNNs are utilized to to mining instruction semantics and structure semantics. However, the authors do not analyze the impact of various GNNs and LLMs.
*  I am not sure if the code property graph and GNNs are really helpful, if replace the GNNs with a simple MLP and other basic GNNs, does the proposed method still work well? And more analysis and details are suggested in the ablation study.

**Questions:**

1. Why do you use GAT? Can we replace it with other GNNs?
2. What is the backbone LLM used in this paper? More details should be included.

**Reviewer Confidence:**

3: The reviewer is confident but not certain that the evaluation is correct

**Scope:**

3: The work is somewhat relevant to the Web and to the track, and is of narrow interest to a sub-community